A path aggregation network with deformable convolution for visual object detection

Rao Chengming 1 2
Hu Zunhao 3
Zhao QiMing 2
Shan Min 2
Mao Li 2 wxmaoli@163.com
1 College of Internet of Things Technology, Wuxi Institute of Technology , Wuxi, Jiangsu , China
2 School of Artificial Intelligence and Computer Science, Jiangnan University , Wuxi, Jiangsu , China
3 School of Internet of Things Engineering, Jiangnan University , Wuxi, Jiangsu , China
Coelho Paulo Jorge
Electronic publication date: 2025 Aug 18
Publication date: 2025
Volume: 11
Electronic Location ID: e3083
Received 2025 Feb 15; Accepted 2025 Jul 5
Copyright: © 2025 Rao et al.
Copyright year: 2025
Copyright holder: Rao et al.
License: This is an open access article distributed under the terms of the Creative Commons Attribution License, which permits unrestricted use, distribution, reproduction and adaptation in any medium and for any purpose provided that it is properly attributed. For attribution, the original author(s), title, publication source (PeerJ Computer Science) and either DOI or URL of the article must be cited.
License URL: https://creativecommons.org/licenses/by/4.0/

Keywords: DePAN architecture, Feature fusion, Object detection

Funding: The authors received no funding for this work.

==============================
One of the main challenges encountered in visual object detection is the multi-scale issue. Many approaches have been proposed to tackle this issue. In this article, we propose a novel neck that can perform effective fusion of multi-scale features for a single-stage object detector. This neck, named the deformable convolution and path aggregation network (DePAN), is an integration of a path aggregation network with a deformable convolution block added to the feature fusion branch to improve the flexibility of feature point sampling. The deformable convolution block is implemented by repeated stacking of a deformable convolution cell. The DePAN neck can be plugged in and easily applied to various models for object detection. We apply the proposed neck to the baseline models of Yolov6-N and YOLOV6-T, and test the improved models on COCO2017 and PASCAL VOC2012 datasets, as well as a medical image dataset. The experimental results verify the effectiveness and applicability in real-world object detection.

Introduction

As one of the hottest directions in the field of artificial intelligence, computer vision has achieved vigorous development during the last two decades. Object detection, as one of the most important tasks in computer vision, has also attracted the attention of both academic and industry communities (Zou et al., 2023). It is also the basis of some other computer visual tasks. For example, the front tasks of pedestrian re-recognition, multi-object tracking and tumor recognition are object detection (Joseph et al., 2021). The goal of object detection is to find interesting targets in single images or continuous video frames, including their positions, sizes, and their categories in the images.

Object detection methods generally can be divided into two categories, namely, traditional object detection algorithms based on image processing and those based on deep learning (Zou et al., 2023). The main difference between these two kinds of methods lies in the process of feature extraction. The procedure of a traditional object detection algorithm generally has the following three steps. First, a series of candidate boxes are generated in the image to be detected. Second, the traditional image processing techniques, such as filtering, edge detection and other methods, are used to extract features for the parts of the image in the candidate box. Finally, whether there is an interested object in the candidate box is determined according to the extracted features such as Harr, HOG (Histogram of Oriented Gradient), SIFT (Scale-Invariant Feature Transform), etc., and an appropriate classifier, e.g., support vector machine (SVM) (Kyrkou & Theocharides, 2012) and adaptive boosting (AdaBoost) (Kyrkou & Theocharides, 2011), is employed to classify the object if there is an interested object in the box area.

The above traditional object detection algorithms have the following disadvantages. First, there are a large amount of redundancy in the generation of the candidate box areas, and it is difficult to determine their shapes and accurately frame the targets of interest. Second, the effect of feature extraction is poor and the extracted feature cannot be used for effective classification. However, these problems have been effectively solved by deep learning. With the continuous maturity of deep learning methods, many deep learning-based object detection algorithms has been developed (Zhao et al., 2019). At present, the mainstream object detection algorithms based on deep learning can be divided into two categories. The first class is the two-stage detection methods represented by fast region-based convolutional neural network (Fast-RCNN) (Girshick, 2015; Ren et al., 2015; Li et al., 2017). In Fast-RCNN, the candidate regions are obtained by the region proposal network in the first stage, and then the convolutional neural network (CNN) is used for bounding box regression and classification of these target regions in the second stage. This method has high detection accuracy but low computational efficiency. The second kind of deep learning-based detection methods are single-stage methods, including You Only Live Once (YOLO) series (Redmon & Divvala, 2016; Wang, Bochkovskiy & Liao, 2023; Jiang et al., 2022), single shot multibox detector (SSD) (Liu et al., 2016; Zhang et al., 2018; Fu et al., 2017) series, and RetinaNet (Lin et al., 2017; Li & Ren, 1905). In a single-stage detection method, the object in the bounding box is classified and its location is regressed directly after feature ex-traction. Such a kind of methods are characterized by fast detection speed with the de-tection accuracy not very high. With the continuous improvement of these methods, the current one-stage algorithms have both faster detection speed and higher accuracy than most of the two-stage algorithms.

Under normal circumstances, there are multiple objects in the image in the object detection task and the object interested is not necessarily a significant object in the image. For example, in the autonomous driving task, it is required to detect the vehicles and pedestrians in the image simultaneously. However, it is often difficult for the network to accurately detect the objects with large differences in sizes. In addition, due to the shooting angle and distance, the objects of the same category to be tested often present different sizes in different images. This is known as the multi-scale issue in the detection task (Pang et al., 2020). How to deal with the multi-scale issue has become the key to the objection detection task.

Currently, there are three measures to tackle these problems. Data pre-processing before model training is widely used technique for tackling multi-scale problems. A typical of such methods is the image pyramid structure (Pang et al., 2019), in which a collection of images of different resolutions are generated from an image and input to the model. Although the image pyramid can alleviate the multi-scale problem to some extent, it significantly increasing the time and space complexities for model training. Another measure for the multi-scale problem is modification of the convolution operation, with deformable convolution being an example of such methods (Dai et al., 2017). The deformable convo-lution is to train an offset so that it may be able to adaptively adjust the sampling position in the image to extract more accurate features. The third strategy is to employer deeper network structures, by which the feature maps generated can carry more semantic in-formation and details of the image. The previous detector such as YOLOv1 (Redmon & Divvala, 2016) only use a single-scale feature map for detection, which results in the poor performance of the net-work in detection of small objects. In YOLOv3 (Redmon & Farhadi, 2018), feature pyramid network (FPN) structure (Lin et al., 2017) is used as a neck to fuse multi-scale features and thus its performance in detection is greatly improved. Subsequently proposed necks, such as spatial pyramid pooling (SPP) (He et al., 2015) and path aggregation network (PAN) (Liu et al., 2018), have been proposed to effectively solve the multi-scale problems existing in object detection.

This work aims to tackle the multi-scale problem by improving the network structure. To this end, we propose in this article a DePAN neck, which combines deformable convolution and RepVGG structure, to fuse the multi-scale feature maps for object detection. Specifically, in the process of object detection, the DePAN neck receives the multi-scale features of the image extracted by the backbone network, and the feature maps of different scales are fused both from top to bottom and from bottom to up. The fused features are input to the detection head for bounding box regression and object classification. This structure can be easily plugged into various networks for object detection. We apply the proposed structure to the baseline models of YOLOV6-N and YOLOV6-T, and carry out the experiments on the COCO2017 dataset. Performance comparison with different methods is made to verify the effectiveness of our proposed the DePAN neck. Be-sides, ablation experiments are performed on PASCAL VOC2012 dataset to further verify the effectiveness of the proposed method.

The rest of this article is organized as follows. ‘Materials and Methods’ provides some preliminaries including the methods of single-stage object detection, path aggregation networks and deformable convolution. ‘Results’ presents the proposed neck with deformable convolutions and path aggregation networks; the experimental results are given and analyzed in this section. Finally, the article is concluded in the last section.

Materials and Methods

This work focuses on improving the single-stage method of object detection to alleviate the multi-scale problem. Thus, in this section we give an introduction to the structure of single-stage detector, path aggregation network, and deformable convolution.

The structure of single-stage detector

Although various single-stage object detection algorithms have been proposed, al-most all of them can be abstracted as three parts: backbone, neck and detection head (Wang, Bochkovskiy & Liao, 2023; Jiang et al., 2022; Liu et al., 2016; Zhang et al., 2018). The structure of the detection process is shown in Fig. 1, where the backbone is used to extract the features of different scales in the image, the neck conducts the fusion operation on multi-scale features extracted by the backbone, and the head output the predictions to the bounding box and the category of the object in the image.

Figure 1 The structure of the single-stage object detector.

In the two-stage object detection method, the region proposal network can generate a large number of redundant candidate boxes and thus consume a lot of computational time. In contrast, the single-stage detection algorithm abandons the region proposal stage and implements end-to-end object detection with YOLO series being the representatives. YOLOv1 (Redmon & Divvala, 2016) uses CNN to extract the features of the original image and directly performs bounding box regression and object classification. Compared to the two-stage method, it has a great improvement on computational efficiency, but its detection accuracy has no significant enhancement. In YOLOv2 (Liu et al., 2018), an anchor frame mechanism is introduced and the prediction of the location is no longer performed by blind and brute regression but by the computation of the offset of the pre-given anchor frame to the actual bounding box, which results in great improvement on detection speed and accuracy. YOLOv3 (Redmon & Farhadi, 2018) further adds FPN (Lin et al., 2017) as the neck to fuse multi -scale features for prediction, with the speed and accuracy of the model further enhanced. YOLOv3 once became one of the most widely used models in the industry.

Path aggregation networks

The current mainstream design of necks is to combine the feature maps of different scales. The fusion methods can be the addition of the corresponding elements of the features, such as FPN (Lin et al., 2017), SPP (He et al., 2015), etc., and also can be stitching of channel dimensions, including the path aggregation network structure used in YOLOv4 (Liu et al., 2018; Bochkovskiy, Wang & Liao, 2004). As high-level features often include high-level semantic information of the image, the corresponding feature maps are generally small and thus upper sampling must be performed before they are fuse them with low-level features. On the contrary, low-level features generally contain low-level information of object shapes but have large feature maps, so down sampling must be performed before they are fused with high-level features. Figure 2 illustrates an example of PAN structure. In this example, the inputs of the PAN are the features of four different scales extracted from the backbone on the leftmost side. Here, {P2, P3, P4, P5,} denotes the feature levels generated by upper sampling and feature fusion. Specifically, features of level P4 are obtained by summation of the input feature map of the same scale with the resultant features upper sampled from features of level P5. By adding the features upper sampled from features of P4 to the original feature map of the same scale, the features of P3 can be obtained. Similarly, the features of P2 are generated by adding the features upper sampled from the features of P3 to the original feature map of the same scale. In this way, the top-down feature fusion is implemented. In the framework, {N2, N3, N4, N5} represents the feature levels generated by down sampling and feature fusion. Put it in detail, the features of N2 are the same as those of P2, the features of N3 are resulted from the element-wise addition of the features of P3 to the features down sampled from those of N2, the features of N4 are obtained from fusion of those of P4 with the features down sampled from those of N3, and the features of N5 are generated from fusion of those of P5 with those down sampled from the features of N4. This is the bottom-up feature fusion in the PAN. Thus, the PAN outputs the features maps of four different spatial size (i.e., N2, N3, N4, and N5) which are then input to the head for prediction of the bounding box and object category.

Figure 2 The framework of the PAN neck.

Deformable convolutions

The sampling mode of the traditional convolution is fixed, and thus its receptive field cannot capture sufficient feature information. The method of increasing the receptive field is to stack convolution layers, but too deep convolutional layers often cause the loss of the low-level features and a large number of redundant parameters. The deformable convolution can change the convolutional sampling locations by a convolutional offset, which is learned during the process of network training (Dai et al., 2017). With adaptive adjustment of the sampling location, the deformable convolution can achieve the effect of increasing the receptive field without increasing the number of layers. The structure of deformable convolution is illustrated in Fig. 3. In the figure, the yellow part is the offset field (or offset map) trained during the network training process. The number of channels of the offset field is 2N, where N is the number of sampling points by the convolution kernel. By taking 3 × 3 convolution kernel as an example, the number of sample points is thus 9, that is, N = 9. When deformable convolution is performed, a total of 2N = 18 channel elements are taken from the same locations in the offset field as the current output locations, and they represent the offsets of nine sampling points in the x and y directions, respectively. By the vector addition of the offsets of each point in the x and y directions, the offsets of nine sampling points can be obtained separately.

Figure 3 The structure of deformable convolutions.

For the input feature map X, the sampling point is first obtained by convolution based on a regular grid R, and the output feature map can be generated by weighted summation of the sampling points with the weights W. The regular grid R of the 3 × 3 convolution kernel is given by

(1) R={(−1,−1),(−1,0),⋯,(0,1),(1,1)}

which represents the locations of the central point and sampling points in the other eight directions. For point P0 in the input feature map, the output of convolution is yielded by

(2) y(P0)=∑Pn∈R⁡w(Pn)⋅x(P¯0+Pn).

Let P0 denote the center point in the input feature map, { P1, P2,…, Pn} represent the eight sampling points in the surrounding directions, w(Pn) indicate the weight associated with the sampling point Pn, and x(P0+Pn) denote the input feature value at the position offset by Pn from the center P0. The convolutional output is computed by performing a weighted sum of the feature values from these sampling points around P0. This mechanism enables the extraction of local features from the input feature map and facilitates feature transformation through spatially learnable weight adjustments.

In the deformable convolution, the regular grid R is augmented with offsets {ΔPn|n=1,2,⋯N}, where N=|R|. Thus, the output feature map y by deformable convolution is

(3) y(P0)=∑Pn∈R⁡w(Pn)⋅x(P0+Pn+ΔPn)

Since the offsets are generally fractional, the pixel value of the sampling point x should be obtained via bilinear interpolation, that is, it is determined by the pixel values of the surrounding points, so that the receptive fields can be increased.

The proposed method

In this work, we propose a framework of neck combining deformable convolution and path aggregation network (DePAN) for the single-stage object detection. The structure of DePAN neck is illustrated in Fig. 4. It takes as input the features of different size extracted by the backbone network. For the features of deep layers, DePAN neck first use a convolutional layer to extract features and adjust the feature dimension, and then undertakes upper sampling through conversion convolution; Next DePAN stitches the up-per sampled features with adjacent low-level features in the channel dimension, and then fuses the features through the DeConVBLOCK. This process corresponds to the fusion from top to bottom on the left part of the DePAN neck in Fig. 4. For the features of shallow layers, DePAN neck carries out down sampling and stitches the down sampled features with adjacent high-level features in the channel dimension, and then fuses the stitched features through DeConVBLOCK, and finally outputs the fused features to the head for prediction. This corresponds to the bottom-up fusion on the right part of the DePAN neck in Fig. 4.

Figure 4 The framework of the DePAN neck.

Specifically, with the given input feature maps, for example X1 and X2 and X3, the proposed DePAN neck first obtains the features of the upper layer P3 by letting P3 = X3, and get the features of lower layers P2 and P1 by

(4) P2=g(h(f(P3),x2)),P1=g(h(f(P2),x1))

where f(∙) is to perform convolution and conversion convolution on the input in turn, h(∙,∙) is to stitch the feature maps of the same size in the channel dimension, and g(∙) represents the operation of DeConvBlock on the stitched feature maps.

P1, P2, P3 represent features at distinct hierarchical levels. P3 is the upper-level feature, while P2 and P1 are lower-level features derived through corresponding function computations. For the input feature maps, x3 is used to directly obtain the upper-level feature P3 whereas x2 and x3 are utilized to compute the lower-level features P2 and P1 via function computations.

With the obtained feature maps of P3, P2 and P1, the DePAN neck further gets the features of the largest size on the output layer, N1, by letting N1 = P1, and get the remaining features N2 and N3 by

(5) N2=g(h(S(N1),P2)),N3=g(h(S(N2),P1))

where S(∙) denotes the convolution operation on the input feature map.

In the DePAN neck, DeConvBlock is composed of N stacked DeConv Cell components, whose structure is illustrated in Fig. 5. The branch structure is adopted in the design of the DeConv Cell. The input features go through the branch of the 1 × 1 convolutional layer, the branch of the 3 × 3 deformation convolutional and batch normalization, and the branch of a separate batch normalization. The results of the three branches are added up and the summation is then input to the activation function.

Figure 5 The structure of DeConv Cell.

In the DEPAN NECK, stitching operation is carried out for feature fusion, and in DECONV cell, addition operation is directly performed to fuse the features. Table 1 is the details of parameter configuration in the DePan NECK applied to YOLOv6 (Li et al., 2022).

Table 1 The parameter settings of DePAN neck.

Structure	Parameter settings	
Input: X1, X2, X3	channels = 256, 512, 1,024	
Conv1	in_channels = 1,024, out_channels = 256, kernel_size = 1, stride = 1	
T1	in_channels = 256, out_channels = 256, kernel_size = 2, stride = 2	
DeConvBlock1	in_channels = 768, out_channels = 256, repeat_N = 12	
Conv2	in_channels = 256, out_channels = 128, kernel_size = 1, stride = 1	
T2	in_channels = 128, out_channels = 128, kernel_size = 2, stride = 2	
DeConvBlock2	in_channels = 384, out_channels = 128, repeat_N = 12	
Conv3	in_channels = 128, out_channels = 128, kernel_size = 3, stride = 2	
DeConvBlock3	in_channels = 256, out_channels = 256, repeat_N = 12	
Conv4	in_channels = 256, out_channels = 256, kernel_size = 3, stride = 2	
DeConvBlock4	in_channels = 256, out_channels = 128, kernel_size = 1, stride = 1	

Each DeConvCell contains a convolution operation and a deformable convolution operation. Table 2 lists specific parameter configuration of the DeConvCell with four stacked DeConvBlocks. According to the parameter configuration, the shape information of the feature maps in each layer can be easily tracked.

Table 2 The parameter settings of DeConvCell.

Block	Structure	Parameter settings of DeConvCell	
DeConvBlock1	Convolution	The first layer: in_channels = 768, out_channels = 256, kernel_size = 3, stride = 1	
The other layers: in_channels = 256, out_channels = 256, kernel_size = 3, stride = 1	
Deformable convolution	The first layer in_channels = 768, out_channels = 256, kernel_size = 3, stride = 1	
The other layers: in_channels = 256, out_channels = 256, kernel_size = 3, stride = 1	
DeConvBlock2	Convolution	The first layer: in_channels = 384, out_channels = 128, kernel_size = 3, stride = 1	
The other layers: in_channels = 128, out_channels = 128, kernel_size = 3, stride = 1	
Deformable convolution	The first layer: in_channels = 384, out_channels = 128, kernel_size = 3, stride = 1	
The other layers: in_channels = 128, out_channels = 128, kernel_size = 3, stride = 1	
DeConvBlock3	Convolution	in_channels = 256, out_channels = 256, kernel_size = 3, stride = 1	
Deformable convolution	in_channels = 256, out_channels = 256, kernel_size = 3, stride = 1	
DeConvBlock4	Convolution	in_channels = 512, out_channels = 512, kernel_size = 3, stride = 1	
Deformable convolution	in_channels = 512, out_channels = 512, kernel_size = 3, stride = 1	

The dimensions need to be adjusted in DeConvBlock1 and DeConvBlock 2, and the output dimensions of P2 and P1 are 256 and 128, respectively. In each DeConvBlock with 12 layers of stacked DeConvCells, the first layer of DeConvCell is responsible for adjusting the dimension with the output dimension of convolution and deformable convolution being the dimension of the expected output feature map; the convolution and de-formable convolution in the remaining 11 layers of DeConvCell do not change the dimension. The 12 layers of DeConvCell in both DeConvBlock3 and DeConvBlock 4 do not change the channel dimensions. The dimensions of the final feature maps of the three different scales output by the DePAN neck, i.e., N1, N2, and N3, are 128, 256 and 512, respectively. These feature maps are then input to the detection head. For different object detection network models, DePAN neck can be plugged and played and is convenient to add to the models, only needing to adjust the channel dimension slightly according to the actual situation.

Results

Experimental settings

To evaluate the effectiveness of the proposed DePAN neck in object detection, we applied the DePAN neck in the YOLOv6 models of two volumes and tested on the CO-CO2017 (Lin et al., 2014) and PASCAL VOC2012 (Everingham et al., 2009) datasets, with the backbone, the head and the loss function the same as those of the original YOLOv6.

COCO2017 is a large dataset provided by Microsoft and contains a large number of labelled data for object detection, segmentation, posture estimation, behavior recognition and other tasks. The labeling information is stored in jason format. The dataset totally contains 118,278 images for training, 5,000 images for verification, and 40,670 images for testing. There are a total of 80 categories in the dataset and each image includes 3.5 objects on average. It is the most widely used dataset in the field of computer vision.

PASCAL VOC2012 is also one of the datasets commonly used in the field of computer vision. It contains 20 object categories, including the labeling information for various visual tasks such as target detection, segmentation, behavior recognition, and the label format is XML. For object detection tasks, PASCAL VOC2012 provides 5,717 images for training and 5,823 ones for verification; for segmentation tasks, the dataset offers 1,464 images for training and 1,449 ones for verification, along with the segmentation results.

All the experiments were run on a workstation containing four Tesla T4 graphics cards. The optimizer used in the model training was the stochastic gradient descent (SGD) algorithm. The size of the input image and the data augmentation was the same as those in the original YOLOv6. Batch size was set to 128 in all the experiments. Since the batch size is 256 in the original YOLOv6, the learning rate was reduced to 1/2 of the original setting, namely 0.05. The cosine scheduler was employed for the learning rate. Each iteration executed for 300 epochs and no pre-training weights were used. The experiments were implemented based on the PyTorch deep learning framework.

Experimental results for different models

We first used our proposed feature fusion framework, the DePAN neck, to the two YOLOv6 models of different parameter volumes, i.e., YOLOv6-N and YOLOv6-T to verify its effectiveness. Performance comparison among different YOLO models were made and the results are listed in Table 3, where the results in bold indicate the higher one between the accuracies of the improved model and the original model.

Table 3 The comparison in detection accuracies among all the compared models.

Models	mAP0.5:0.95(%)	mAP0.5(%)	Parameters	FLOPs	
YOLOv5-N	28.0	45.7	1.9 M	4.5 G	
YOLOv5-S	37.4	56.8	7.2 M	16.5 G	
YOLOX-Tiny (Ge et al., 2021)	32.8	50.3	5.1 M	6.5 G	
YOLOX-S (Ge et al., 2021)	40.5	59.3	9.0 M	26.8 G	
PPYOLOE-S (Xu et al., 2022)	43.1	59.6	7.9 M	27.4 G	
YOLOv7-Tiny (Wang, Bochkovskiy & Liao, 2022)	33.3	49.9	6.2 M	5.8 G	
YOLOv8-N (Yolov8, 2023)	37.3	52.6	3.2	8.7	
YOLOv8-S (Yolov8, 2023)	44.9	61.8	11.2	28.6	
YOLOv9-S (Wang, Yeh & Mark Liao, 2024)	46.8	63.4	7.2	26.7	
YOLOv10-N (Wang et al., 2024)	38.5	53.8	53.8	6.7	
YOLOv10-S (Wang et al., 2024)	46.3	63.0	7.2	21.6	
YOLO11-N (yolov11, 2024)	39.5	55.3	2.6	6.5	
YOLO11-S (yolov11, 2024)	47.0	46.9	9.4	21.5	
YOLOv12-N (Tian, Ye & Doermann, 2025)	40.6	56.7	2.6	6.5	
YOLOv12-S (Tian, Ye & Doermann, 2025)	48.0	65.0	9.3	21.4	
RT-DETR-R50 (Zhao et al., 2024)	53.1	71.3	42.0	136.0	
RT-DETRv2-R50
(Lv et al., 2024)	53.4	71.6	42.0	136.0	
YOLOv6-N (Li et al., 2022)	35.9	51.2	4.3 M	11.1 G	
YOLOv6-T (Li et al., 2022)	40.3	56.6	15.0 M	36.7 G	
YOLOv6-N*	36.5	52.4	3.8 M	10.6 G	
YOLOv6-T*	41.1	58.0	11.2 M	28.0 G	
Note:

An asterisk (*) indicates a model incorporating the proposed method.

It can be seen from Table 3 that the detection accuracies by the baseline models, Yolov6-N and YOLOv6-T, on COCO2017 dataset are 35.9% and 40.3% of mAP0.5:0.95, and 51.2%, and 56.6% of mAP0.5. Yolov6-N* and YOLOV6-T* denote the YOLOv6-N and YOLOv6-T, whose necks are replaced by the proposed DePAN necks respectively. Thanks to the enhancement of feature fusion capabilities by the proposed methods, the model performance was significantly improved. Specifically, for the YOLOv6-N*, its mAP0.5:0.95 increased by 0.7% from 35.9% to 36.5%, and its mAP0.5 increased by 1.2% from 51.2% to 52.4%. For YOLOv6-T*, its mAP0.5:0.95 increased by 0.8% from 40.3% to 41.1%, and its mAP0.5 increased by 1.4% from 56.6% to 58.0%. Finally, a comparison was also made with the latest version of the YOLO series. There is not a big gap in mAP accuracy. And because the detector of the DETR series has a large number of parameters, although the operation speed of our detector is less, the effect comparison is not very ideal.

In addition, in order to verify the effectiveness of the proposed method in the detection of small objects, we also tested YOLOv6-N* and YOLOv6-T*, which adopts our proposed DePAN necks, on data of small objects in COCO2017. The results for mAP of the models are shown in Table 4. It is shown that compared to the baseline models YOLov6-N and YOLOV6-T, YOLOv6-N* and YOLOv6-T*, with the help of our DePAN neck, obtained higher accuracies in small object detection. Specifically, their mAP0.5:0.95 increase by 0.5% from 16.5 to 17.0 and by 1.3% from 20.3% to 21.6%, respectively.

Table 4 The results of accuracies in small object detection on COCO2017 dataset.

Models	mAP0.5:0.95(%)	Parameters	FLOPs	
YOLOv6-N (Li et al., 2022)	16.5	4.3 M	11.1 G	
YOLOv6-T (Li et al., 2022)	20.3	15.0 M	36.7 G	
YOLOv6-N*	17.0	3.8 M	10.6 G	
YOLOv6-T*	21.6	11.2 M	28.0 G	
Note:

An asterisk (*) indicates a model incorporating the proposed method.

Ablation studies

In order to verify the effectiveness and generalizability of our proposed DePAN neck, we undertook the ablation studies on the number of stacked DeConvCells in the DeConvBlock with another dataset, PASCAL VOC2012.

We tested YOLOv6-N* and YOLOv6-T* and their baseline models on PASCAL VOC2012 dataset, training the models on the training set and verifying them on the verification set. The results are reported in Table 5. As can be seen, Yolov6-N*and YOLOV6-T* both had remarkable improvement on detection accuracies over their corresponding baseline models respectively. For YOLOv6-N*, it had an increase of 1.3% in mAP0.5:0.95 from 47.1% to 48.4% and an increase of 0.9% in mAP0.5 from 66.7% to 67.6%. For YOLOv6-T*, it had an increase of 1.1% in mAP0.5:0.95 from 49.1% to 50.2% and an increase of 0.9% from 49.1% to 50.2%, and an increase of 2.3% in mAP0.5 from 67.3% to 69.6%. We integrated the module on YOLO11-N. Compared with the detectors after the sixth generation, the mAP0.5:0.95 increased from 40.9% to 46.5%, with a significant improvement in effect. From these results, we can see that the proposed method can also effectively improve the model performance in object detection on different datasets, indicating its good generalizability.

Table 5 The detection results of the models on PASCAL VOC2012 dataset.

Models	mAP0.5:0.95(%)	mAP0.5(%)	Parameters	FLOPs	
YOLOv6-N (Li et al., 2022)	47.1	66.7	4.3 M	11.1 G	
YOLOv6-T (Li et al., 2022)	49.1	67.3	15.0 M	36.7 G	
YOLOv6-N*	48.4	67.6	3.8 M	10.6 G	
YOLOv6-T*	50.2	69.6	11.2 M	28.0 G	
YOLOv8-N (Yolov8, 2023)	40.9	58.8	3.01 M	8.1 G	
YOLOv9-tiny (Wang, Yeh & Mark Liao, 2024)	37.9	71.3	2.62 M	10.7 G	
YOLOv10n (Wang et al., 2024)	36.8	64.3	2.69	6.7 G	
YOLO11-N (yolov11, 2024)	45.8	64.6	2.6 M	6.3 G	
YOLO11-N*	46.5	64.9	2.5 M	6.2 G	
Note:

An asterisk (*) indicates models which used our proposed DePAN necks.

The DePAN neck proposed in this work has DeConvBlocks, each of which consists of repeatedly stacked DeConv cells. In order to investigate how the number of stacked DeConv cells influences the performance of the DePAN neck, we performed the experiments on PASCAL VOC2012 with four different settings for the number of cells. The results are shown in Table 6. When the numbers of the stacked cells in the four DeConvBlocks were 12, 12, 12 and 12, respectively, YOLOv6-N*obtained a mAP0.5:0.95 of 48.4% and YOLOv6-T* 50.2% respectively. The results from this setting of the numbers of stacked cells are better than the results with the numbers set to 6, 6, 12, and 12 and the results with the number set to 12, 12, 6, and 6. It is shown that when the numbers of the stacked cells were set to 16, 16, 16 and 16 respectively, both of the models showed only slightly better performance than the model performance with 12, 12, 12 and 12 stacked cells in the four DeConvBlocks respectively. Therefore, in the other experiments in this work, we set all the numbers of stacked DeConv cells in the four DeConvBlocks to 12. Additionally, we have also conducted a similar experiment and applied it to YOLOv8 and YOLOv9. The relevant experimental comparison results are presented in Table 6. It can be observed that with the addition of DeConv cells, the experimental effect continuously improves, proving that our module is effective.

Table 6 Comparison in model accuracy with different numbers of stacked DeConv cells on PASCAL VOC2012 dataset.

Models	Number of stacked cells	mAP0.5:0.95(%)	mAP0.5(%)	
YOLOv6-N	[6, 6, 12, 12]	47.7	67.0	
[12, 12, 6, 6]	47.8	67.3	
[12, 12, 12, 12]	48.4	67.6	
[16, 16, 16, 16]	48.7	67.6	
YOLOv6-T	[6, 6, 12, 12]	48.8	68.2	
[12, 12, 6, 6]	49.0	68.4	
[12, 12, 12, 12]	50.2	69.6	
[16, 16, 16, 16]	50.3	69.2	
YOLOv8	[8, 8, 16, 16]	51.1	70.5	
[16, 16, 16, 16]	51.3	70.8	
[16, 16, 16, 16]	52.7	71.9	
YOLOv9	12, 12, 24, 24]	53.8	73.4	
[24, 24, 12, 12]	54.0	73.6	
[24, 24, 24, 24]	55.3	74.8	

Experimental results on medical image dataset

In order to further demonstrate the applicability of our proposed method in re-al-word problems, we also performed experiments on medical imaging dataset. The dataset contains 1,076 CT images of breast nodules, including 476 benign cases and 600 malignant ones. The size of each image is 224 * 224. The tumor nodules were marked with DarkLabel labeling software. Benign and malign images together were divided into training set and verification set according to the ratio of 7 to 3. That is, there were 753 images in the training set, including 333 benign images and 420 malignant images, and 333 images in the verification set, including 143 benign images and 180 malignant images.

The experimental results are shown in Table 7, where Yolov6-N*and YOLOV6-T* are the models with the proposed DePAN neck added. It can be seen from the experimental results that on the breast CT image dataset, the mAP0.5:0.95 obtained by YOLOv6-N* is 43.3%, higher than that of its baseline (41.9%) by 1.4%; the mAP0.5 obtained by YOLOv6-N* is higher than that of its baseline by 4.2%. For YOLOv6-T*, the mAP0.5:0.95 increased by 1.1% from 43.4% to 44.5% and the mAP0.5 increased by 1.0% from 84.3% to 85.3%. This means that the proposed DePAN neck can improve the model performance in tumor detection on medical images.

Table 7 The effect in tumor detection on the medical image dataset.

Models	mAP0.5:0.95(%)	mAP0.5(%)	Parameters	FLOPs	
YOLOv6-N (Li et al., 2022)	41.9	81.8	4.3 M	11.1 G	
YOLOv6-T (Li et al., 2022)	43.4	84.3	15.0 M	36.7 G	
YOLOv6-N*	43.3	82.2	3.8 M	10.6 G	
YOLOv6-T*	44.5	85.3	11.2 M	28.0 G	
Note:

An asterisk (*) indicates models which used our proposed DePAN necks.

Figure 6 gives the results of prediction by YOLov6-T and YOLOV6-T* on the breast CT medical images. Figure 6A are the labelled images of a benign tumor and a malignant tumor respectively. Figure 6B shows the results of prediction to the corresponding tumors by YOLOv6-T, and Fig. 6C provides the results of prediction to the tumors by YOLOv6-T with DePAN neck (i.e., VOLOv6-T*). It can be observed that the benign tumor was classified to be malignant one by YOLOv6 and the bounding box predicted by YOLOv6 for the malignant tumor was too large and not accurate enough. However, by using VOLOv6-T with the DePAN neck, the malignant tumor was classified correctly and the predicted bounding box was more accurate.

Figure 6 The prediction outcome on breast CT images.

Figures 7 and 8 illustrate the prediction results on the PASCAL VOC2012 dataset. As can be observed, the predicted classes are highly accurate, and the bounding boxes effectively encapsulate the detected objects. The visualizations demonstrate the robustness and precision of our model in identifying and localizing various objects within the images. The clear delineation of the bounding boxes and the correct classification of the objects underscore the effectiveness of our approach in real-world scenarios, highlighting its potential for practical applications in object detection tasks.

Figure 7 The performance of our model on the VOC dataset.

Figure 8 The performance of our model on the COCO dataset.

Conclusions

To perform multi-scale feature fusion effectively in the object detection tasks, this article proposed a DePAN neck based on the deformable convolution and the path aggregation network. It can be easily used to in various object detection models to enhance the effect of the multi-scale feature fusion. In the framework of the DePAN, in terms of structural design, refer to the PAN structure, and uses DeConvBlock with deformed convolution to solve the problem of receptive field of the tradition traditional convolution, and enhance the feature extraction ability of different scale feature diagrams. The main experiments and rich melting experiments are designed on Pascal VOC2012 and COCO2017 data sets to illustrate the effectiveness of the DEPAN NECK structure pro-posed in this chapter. The experimental results were analyzed visualized to show the effectiveness of the method more intuitively.

Additional Information and Declarations

Competing Interests

The authors declare that they have no competing interests.

Author Contributions

Chengming Rao conceived and designed the experiments, performed the experiments, analyzed the data, performed the computation work, prepared figures and/or tables, and approved the final draft.

Zunhao Hu conceived and designed the experiments, performed the experiments, analyzed the data, performed the computation work, prepared figures and/or tables, authored or reviewed drafts of the article, and approved the final draft.

QiMing Zhao conceived and designed the experiments, performed the experiments, performed the computation work, prepared figures and/or tables, and approved the final draft.

Min Shan analyzed the data, performed the computation work, prepared figures and/or tables, authored or reviewed drafts of the article, and approved the final draft.

Li Mao analyzed the data, authored or reviewed drafts of the article, and approved the final draft.

Data Availability

The following information was supplied regarding data availability:

The data is available at: https://cocodataset.org.

https://www.kaggle.com/datasets/gopalbhattrai/pascal-voc-2012-dataset.

Our code is available at Zenodo: Rao. (2025). A Path Aggregation Network with Deformable Convolution for Visual Object Detection. Zenodo. https://doi.org/10.5281/zenodo.14613602.

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
