# Peer review of "A path aggregation network with deformable convolution for visual object detection"

_PeerJ Computer Science, doi:10.7717/peerj-cs.3083_

## Round 0.1 · original submission · Major Revisions

**Language Note:** The review process has identified that the English language must be improved. PeerJ can provide language editing services - please contact us at [email protected] for pricing (be sure to provide your manuscript number and title). Alternatively, you should make your own arrangements to improve the language quality and provide details in your response letter. – PeerJ Staff

Reviewer 1 ·

Basic reporting

This paper proposes a DePAN neck, which combines deformable convolution and RepVGG structure, to fuse the multi-scale feature maps for object detection. There is still some work that needs to be polished. Here are the details of my suggestions:
(1) Fig. 1 and Fig. 2: The titles of these two figures are exactly the same. Fig. 2 is an example of PAN structure.
(2) Lines 185-188 and Lines 209-215: Explain the meaning of symbols in the formula.
(3)The paper should discuss whether the proposed DePAN is universal. Can it be integrated into other versions of Yolo? What is the effect of combining different versions of Yolo?
(4)Table 3: There are relatively few comparative experiments, only the YOLO model, and the new YOLOv8-YOLOv10 models have not been included in the comparative experiments; The paper should also include comparisons of models such as DETR. The paper can refer to some references on the object detection model DETR, such as doi:10.1007/s11554-025-01632-y
(5) This paper emphasizes multi-scale object detection, but there is no detailed comparison of the detection results of the proposed model for large, medium, and small objects in the experiment.
(6) The resolution of the images in the paper is not high, and the module color matching in the model can refer to other papers.
(7) Please share the code of the paper in the Github.

Experimental design

No comment.

Validity of the findings

No comment.

Additional comments

No comment.

Reviewer 2 ·

Basic reporting

The manuscript addresses the challenge of multi-scale object recognition in visual object detection by proposing a novel neck network designed for effective multi-scale feature fusion. The proposed network, DePAN, integrates Path Aggregation Network (PAN) with deformable convolutional blocks to enhance the flexibility of feature point sampling. Experimental evaluations conducted on datasets such as COCO2017, PASCAL VOC2012, and a medical image dataset demonstrate promising results, indicating the method's effectiveness. Additionally, the paper is well-written, presenting the methodology and results in a clear and structured manner.

Experimental design

While the authors have conducted experiments to validate the performance of DePAN, the comparative analysis is insufficient. The study primarily compares DePAN-enhanced YOLOv6 models against their baseline counterparts. However, it lacks comparisons with more recent and state-of-the-art object detection methods. Given that DePAN is designed as a plug-and-play module, its applicability and effectiveness should be tested across a broader range of models, including newer versions such as YOLOv8, YOLOv9, YOLOv10, and Hyper-YOLO. Expanding the experimental scope in this manner would provide a more comprehensive evaluation of DePAN's performance and generalizability.

Validity of the findings

The experimental results presented in Table 3 raise concerns regarding the validity and impact of the proposed method. Firstly, the performance improvement over YOLOv6 is marginal, which may not justify the added complexity introduced by DePAN. Secondly, when compared to the PPYOLOE method, DePAN not only underperforms but also requires a higher number of parameters, suggesting that the proposed approach may not be as efficient or effective as existing advanced methods. Furthermore, Table 6 indicates that increasing the number of stacked deformable convolutional (DeConv) blocks leads to performance gains. This observation prompts the question of whether further stacking could yield even better results, and whether there is an optimal number of DeConv blocks beyond which performance gains plateau or diminish.

Additional comments

1. Visualization: The manuscript would benefit from the inclusion of more visualizations, such as feature maps or detection examples, to illustrate the qualitative improvements brought by DePAN. Visual aids can significantly enhance the reader's understanding of how DePAN affects feature fusion and object detection performance.
2. Theoretical Analysis: The paper lacks a thorough theoretical analysis of the proposed method. Specifically, there is a need for a comparative discussion with other multi-scale aggregation techniques, such as Hyper-YOLO. Providing a theoretical framework or rationale for why DePAN outperforms or differs from existing methods would strengthen the manuscript.

---

## Round 0.2 · accepted · Accept

Dear authors, we are pleased to verify that you meet the reviewer's valuable feedback to improve your research.

Thank you for considering PeerJ Computer Science and submitting your work.

Kind regards
PCoelho

Reviewer 1 ·

Basic reporting

-

Experimental design

-

Validity of the findings

-